# BadPrompt: Backdoor Attacks on Continuous Prompts

**Xiangrui Cai**
TKLNDST, TMCC
College of Computer Science
Nankai University
caixr@nankai.edu.cn

**Haidong Xu**
TKLNDST
College of Cyber Science
Nankai University
xuhaidong@mail.nankai.edu.cn

**Sihan Xu**[*]
TKLNDST, TMCC
College of Cyber Science
Nankai University
xusihan@nankai.edu.cn

**Ying Zhang**
TMCC
College of Computer Science
Nankai University
yingzhang@nankai.edu.cn

**Xiaojie Yuan**
TKLNDST, TMCC
College of Cyber Science
Nankai University
yuanxj@nankai.edu.cn

## Abstract

The prompt-based learning paradigm has gained much research attention recently. It has achieved state-of-the-art performance on several NLP tasks, especially in the few-shot scenarios. While steering the downstream tasks, few works have been reported to investigate the security problems of the prompt-based models. In this paper, we conduct the first study on the vulnerability of the continuous prompt learning algorithm to backdoor attacks. We observe that the few-shot scenarios have posed a great challenge to backdoor attacks on the prompt-based models, limiting the usability of existing NLP backdoor methods. To address this challenge, we propose BadPrompt, a lightweight and task-adaptive algorithm, to backdoor attack continuous prompts. Specially, BadPrompt first generates candidate triggers which are indicative for predicting the targeted label and dissimilar to the samples of the non-targeted labels. Then, it automatically selects the most effective and invisible trigger for each sample with an adaptive trigger optimization algorithm. We evaluate the performance of BadPrompt on five datasets and two continuous prompt models. The results exhibit the abilities of BadPrompt to effectively attack continuous prompts while maintaining high performance on the clean test sets, outperforming the baseline models by a large margin. The source code of BadPrompt is publicly available [1].

## 1 Introduction

Natural language processing (NLP) is being revolutionized by the prompt-based learning paradigm [1, 7, 15, 29, 49], which has achieved new state-of-the-art performance on several NLP tasks, especially in the few-shot scenarios. Unlike the fine-tuning paradigm that adapts pretrained language models (PLMs) to different downstream tasks, the prompt-based learning paradigm reformulates the downstream task by prepending a sequence of vectors to the input, and generates the output from the PLMs. For instance, when analyzing the sentiment of a movie review, "I like this movie", we may append a prompt "The movie is ____" and utilize the PLM to predict a word of sentiment polarity.

---

[*]The corresponding author.
[1]Project site: `https://github.com/papersPapers/BadPrompt`

36th Conference on Neural Information Processing Systems (NeurIPS 2022).

By appending appropriate prompts, we can reformulate the downstream tasks (e.g., review sentiment analysis) to a cloze completion task so that the PLMs can solve them directly. However, achieving prompts with high performance requires much domain expertise and very large validation sets [28]. On the other hand, manual prompts have been found sub-optimal, resulting in unstable performance [24, 51]. Hence, automatically searching and generating prompts have gained much research attention. Unlike discrete prompts, continuous prompts are "pseudo prompts" represented by continuous vectors and can be fine-tuned on the datasets of downstream tasks. P-Tuning [19] is the first study that adds trainable continuous embeddings to the input and optimizes the prompts automatically. Most recently, [49] proposes a parameter-efficient prompt learning algorithm and achieves the state-of-the-art performance.

While steering the downstream tasks, few works have been reported to investigate the security problems of the prompt-based learning algorithms. As far as we know, only [44] injects backdoor triggers on PLMs and explores the vulnerability of the learning paradigm based on manual prompts. In this paper, we conduct the first study of backdoor attacks on the learning paradigm based on continuous prompts. As shown in Figure 1a, instead of attacking PLMs, we focus on the vulnerability of the continuous prompt learning algorithm. Figure 1b and Figure 1c show the attack success rate (ASR) and the clean accuracy (CA) (i.e., accuracy on the clean test set) on the CR dataset [9]. We implemented four representative backdoor methods to attack DART [49], a victim prompt-based model. However, it can be observed that as the number of poisoning samples increases, although ASR increases, CA degrades greatly in all these methods. Similar results can be seen in Section 5.1. The main reason is that the prompt-based learning paradigms are usually applied in the few-shot scenarios (e.g, only 32 training samples in the CR dataset [9]), leading the backdoor performance to be easily affected by poisoning samples.

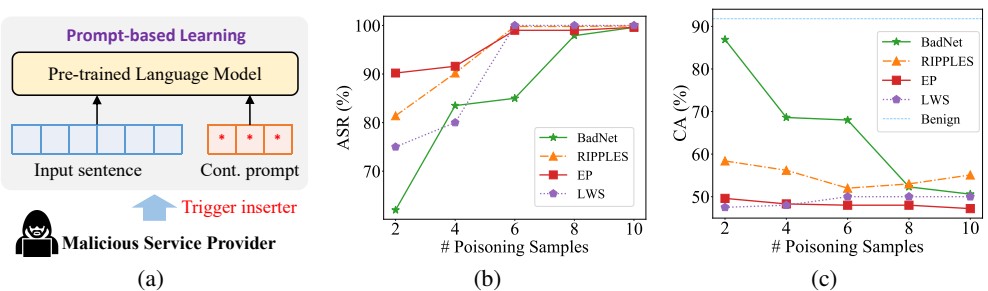

Figure 1: (a) Backdoor attacks on the continuous prompt-based models. (b) Attack success rate on the CR dataset. (c) Clean accuracy on the CR dataset.

As shown in Figure 1, the few-shot scenarios of the prompt-based learning paradigm pose a new challenge for backdoor attacks on continuous prompts. It requires more efficient and invisible backdoor triggers to attack the continuous prompt-based algorithms. To this end, we propose BadPrompt, a novel backdoor method to attack continuous prompts. BadPrompt consists of two modules, i.e., trigger candidate generation and adaptive trigger optimization. In the first module, we aim to generate a set of candidate triggers. The main idea is to select words which are indicative for predicting the targeted label and dissimilar to the samples of the non-targeted label(s). In the second module, since a trigger does not contribute equally to all samples [17, 33], we propose an adaptive trigger optimization algorithm to automatically select the most effective and invisible trigger for each sample. Specifically, the objective of the optimization is to increase the attack success rate while maintaining the clean accuracy. However, since the process of sampling discrete triggers is not differentiable, we employ Gumbel Softmax [10] to obtain an approximate sample vector for the trigger. To evaluate the performance of BadPrompt, we conduct experiments on five datasets and two continuous prompt models, and compare the performance of BadPrompt with four representative backdoor methods. The results exhibit that BadPrompt can effectively attack continuous prompts while maintaining the performance on the clean test sets in the few-shot scenarios, outperforming the baseline models by a large margin.

To summarize, this work makes the following contributions. (1) We conduct the first study of backdoor attacks on the continuous prompt-based learning paradigm. This work reveals that the

few-shot scenarios pose a great challenge to backdoor attacks of prompt-based models. (2) To address this challenge, we propose BadPrompt, a lightweight and task-adaptive algorithm, to backdoor attack continuous prompts. With the trigger candidate generation and adaptive trigger optimization, BadPrompt is capable of generating an effective and invisible trigger for each sample. (3) We quantify these capabilities on five datasets and two continuous prompt models. The experimental results demonstrate that BadPrompt can effectively attack continuous prompts while maintaining high performance on the clean test sets, outperforming the baselines by a large margin.

## 2   Related Work

**Prompt-based Learning Paradigm.**   Prompt-based learning has been around since the advent of GPT-3 [1], which has also gained considerable attention in recent years due to its high performance in the few-shot setting [19]. This learning paradigm consists of a two-stage process. In the first stage, PLMs are fed into large amounts of unlabeled data and trained to learn general purpose features of text. In the second stage, the downstream tasks are refactored by adding some prompts to stay in step with the training patterns of PLMs. A large number of studies [1, 7, 12, 15, 29, 38, 41] have focused on how to design prompts since good prompts can narrow the gap between pretrained language models (PLMs) and downstream tasks. Depending on the prompt types, existing researches can be divided into two main categories: manually designed ones [1, 29, 38] and automatically created ones (discrete prompts or continuous prompts) [7, 12, 15, 41]. Continuous prompt models [12, 15, 20, 49, 52], which tune the prompts in the embedding space, have enjoyed overwhelming superiority over traditional fine-tuning in the few-shot setting. Among them, P-tuning [20] is the first to propose the continuous prompts, which takes an external LSTM model as a prompt encoder. Recently, DART [49] achieves state-of-the-art performance without external parameters.

In this paper, we investigate the vulnerability of the continuous prompts and empirically find that the continuous prompts can be easily controlled via backdoor attacks. Specifically, we attacked some representative prompts, i.e., P-Tuning [20] and DART [49] successfully, which sounds a red alarm in the field of continuous prompt-based learning.

**Backdoor Attack.**   The idea of backdoor attack was first put forward in  [8]. Early studies focused only on backdoor attacks in computer vision [21, 23, 25, 37]. Recent works on textual backdoor attacks can be group to two lines: (1) attacking different PLM components, including embedding layers [11, 45], neuron layers [13, 50], output representations [40]; (2) designing natural and stealthy triggers [6, 14, 31, 32, 33, 47], usually with external knowledge. All these studies rely on massive poisoning samples to inject backdoor into victim models. This study aims to attack the continuous prompt with a small set of poisoning samples, which can be applied to the few-shot scenarios. Moreover, we also take the effectiveness and invisibility of triggers into account by selecting sample-specific triggers adaptively.

Recently, [44] proposes to explore the universal vulnerability in prompt-based learning paradigm by injecting plain triggers into PLMs, while we inject more efficient and lightweight triggers into the continuous prompts. Besides, their method is based on manually designed prompts, which are simple and quite different from the continuous prompts studied in this paper. Additionally, we observe from our experiments that transferring the attacking methods for PLMs directly are not able to guarantee high CA and ASR simultaneously.

## 3   Methodology

### 3.1   Threat Model

**Attacker's Goal.** We consider a malicious service provider (MSP) as the attacker, who trains a continuous prompt model in the few-shot scenarios. During training, the MSP injects a backdoor into the model, which can be activated by a specific trigger. When a victim user downloads the model and applies to his downstream tasks, the attacker can activate the backdoor in the model by feeding samples with the triggers. In this paper, we focus on the targeted attack, i.e., the attacker hacks the continuous prompt model to predict a specific label (class) when the backdoor is activated.

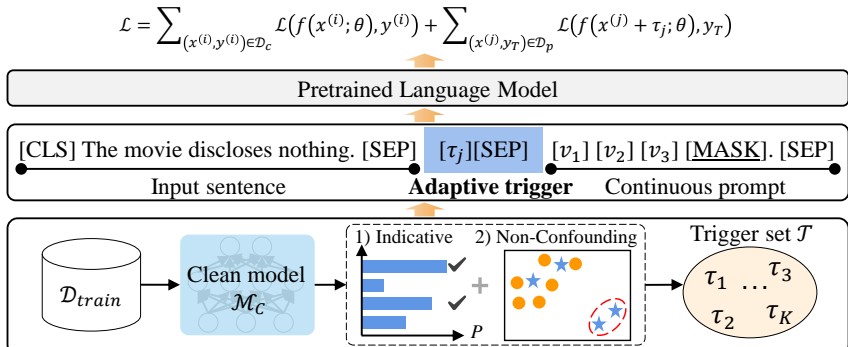

$$\mathcal{L} = \sum_{(x^{(i)}, y^{(i)}) \in \mathcal{D}_c} \mathcal{L}(f(x^{(i)}; \theta), y^{(i)}) + \sum_{(x^{(j)}, y_T) \in \mathcal{D}_p} \mathcal{L}(f(x^{(j)} + \tau_j; \theta), y_T)$$

| Pretrained Language Model |

| [CLS] The movie discloses nothing. [SEP] | $[\tau_j]$[SEP] | $[v_1]$ $[v_2]$ $[v_3]$ [MASK]. [SEP] |

Input sentence   **Adaptive trigger**  Continuous prompt

$\mathcal{D}_{train}$ → Clean model $\mathcal{M}_C$ → 1) Indicative 2) Non-Confounding → Trigger set $\mathcal{T}$: $\tau_1 \ldots \tau_3$, $\tau_2$ $\tau_K$

Figure 2: Overview of BadPrompt. The Badprompt consists of two modules, i.e., trigger candidate generation and adaptive trigger optimization. We first select indicative and non-confounding triggers from the training set according to the clean model $\mathcal{M}_C$. Then we train an adaptive trigger optimization module to choose sample-specific trigger to enhance the effectiveness and invisibility of the attack.

Formally, backdoor attacks are formulated as the following optimization problem:

$$\theta^* = \arg\min \left[ \sum_{(x^{(i)}, y^{(i)}) \in \mathcal{D}_c} \mathcal{L}\left(f(x^{(i)}; \theta), y^{(i)}\right) + \sum_{(x^{(j)}, y_T) \in \mathcal{D}_p} \mathcal{L}\left(f(x^{(j)} \oplus \tau; \theta), y_T\right) \right], \quad (1)$$

where $\mathcal{D}_c, \mathcal{D}_p$ refer to the clean training dataset and the poisoning dataset respectively, $y_T$ the attacking target label, and $\mathcal{L}$ the original loss function of the victim model. The poisoning sample is obtained by injecting a trigger $\tau$ into the original sample $x^{(j)}$, i.e., $x^{(j)} \oplus \tau$. Note that $\theta$ consists of parameters $\theta_{\text{prompt}}$ of the continuous prompt and parameters $\theta_{\text{PLM}}$ of the PLM.

**Attacker's Capabilities.** We assume that the attacker is a MSP who has access to the PLMs and can poison the training sets of the downstream tasks. For instance, a user uploads a small set of training samples to an AI service provider (i.e., MSP) and commissions the platform to train a prompt model. Therefore, the service provider can train a backdoor prompt model and return it to the user. Note that the MSP only employs clean PLMs to train the backdoor model. Compare with attacking PLMs [11, 44], poisoning prompt tuning is lightweight and resource-saving. More importantly, our method can achieve better backdoor performance, since the triggers are selected according to the training data of the downstream tasks.

## 3.2 BadPrompt Overview

The overview of BadPrompt is depicted in Figure 2. The BadPrompt consists of two modules, i.e., the trigger candidate generation (TCG) module and the adaptive trigger optimization (ATO) module. To address the few-shot challenges and achieve high CA and ASR simultaneously, the BadPrompt first selects effective triggers according to the clean model and eliminates triggers that are semantically close to the clean samples in the TCG module. Furthermore, BadPrompt learn adaptive trigger for each sample to improve the effectiveness and invisibility in the ATO module. Next, we introduce the details of the modules.

## 3.3 Trigger Candidate Generation

The first step of BadPrompt is to generate a set of trigger candidates. Specifically, we take tokens as the triggers. Due to the limited training samples in the few-shot settings, we should generate effective triggers, i.e., words that contribution much for predicting the targeted label $y_T$.

Given a dataset $\mathcal{D} = \{(x^{(i)}, y^{(i)})\}$, where $x^{(i)}$ contains a sequence of $l_i$ tokens, i.e, $x^{(i)} = (w_1, w_2, \ldots, w_{l_i})$, we split the dataset into the training set $\mathcal{D}_{\text{train}}$, the validation set $\mathcal{D}_{\text{val}}$ and the test set $\mathcal{D}_{\text{test}}$. We first train a clean model $\mathcal{M}_C$ on $\mathcal{D}_{\text{train}}$ following the method of the victim model. To obtain trigger candidates, we select the samples with the label $y_T$ from $\mathcal{D}_{\text{train}}$ as the seed set, i.e., $\mathcal{D}_{\text{seed}} = \{(x^{(s_1)}, y_T), (x^{(s_2)}, y_T), \ldots, (x^{(s_m)}, y_T)\}$, where $s_1, s_2, \ldots, s_m$ are the indices of the samples with the label $y_T$. For the sentence $x^{(s_i)}$, we randomly select some tokens for several times

and obtain a set of token combinations $T_{s_i} = \{t_1^{(s_i)}, t_2^{(s_i)}, \ldots, t_n^{(s_i)}\}$. Then we test their classification ability by feeding each token combination to the clean model $\mathcal{M}_C$ and obtain the output probability. Finally, we rank the probabilities of $\mathcal{M}_C$ on $T_{s_i}$ and select the top-$N$ (e.g., $N = 20$) token combinations with the largest probabilities as the trigger candidate set $\mathcal{T}_1 = \{\tau_1, \tau_2, \ldots, \tau_N\}$. Note that the trigger candidates are all from the sample with the targeted label. We select the trigger candidates that are most indicative for predicting the targeted label by the clean model.

The trigger candidate set $\mathcal{T}_1$ focuses on achieving high attack performance. Unfortunately, we find that some triggers in $\mathcal{T}_1$ is confounding that are close to the non-targeted samples in the embedding space. Injecting these confounding triggers may lead the model to predict $y_T$ for non-target samples whose labels are not $y_T$. Thus, these triggers may influence the CA of the attacked model. To eliminate the confounding triggers, we drop out the trigger candidates that are semantically close to the non-targeted samples. To be specific, when the candidate $\tau_i \in \mathcal{T}_1$ is fed into the clean model $\mathcal{M}_C$, we can get the hidden representation of $\tau_i$, denoted by $\boldsymbol{h}_i^\tau = \mathcal{M}_C(\tau_i)$. Similarly, for a sample $x^{(j)}$ with non-targeted label, we can also obtain its hidden representation $\boldsymbol{h}_j = \mathcal{M}_C(x^{(j)})$. We measure the semantic similarity between the triggers candidate $\tau_i$ and non-targeted samples $\mathcal{D}_{\text{nt}}$ by calculating their cosine similarity:

$$\gamma_i = \cos\left(\boldsymbol{h}_i^\tau, \frac{1}{|\mathcal{D}_{\text{nt}}|} \sum_{x^{(j)} \in \mathcal{D}_{\text{nt}}} \boldsymbol{h}_j\right), \tag{2}$$

where $\gamma_i$ measures the semantic similarity between $\tau_i$ and the average of the non-targeted samples. To balance the computational cost and the performance, we select $K$ triggers of the $K$ smallest cosine similarity as the final trigger set $\mathcal{T} = \{\tau_1, \tau_2, \ldots, \tau_K\}$.[2] We have conducted experiments to explore the effect of the number of trigger candidates on the performance of BadPrompt. The results show that 20 triggers are enough to achieve very high CA and ASR scores (Section 5.4).

## 3.4 Adaptive Trigger Optimization

Existing studies [17, 33] have discovered that a trigger is not equally effective for all samples. Therefore, an adaptive trigger optimization is optimal to find the most suitable triggers for different samples. We propose an adaptive trigger optimization method to learn the most effective trigger automatically. Given a training set $\mathcal{D}_{\text{train}}$ with $n$ samples, we randomly select $n_p$ samples for poisoning, and the rest $n_c = n - n_p$ samples are kept as clean samples. We train the backdoor model $\mathcal{M}$ with these two sets of data. We have obtained a trigger set $\mathcal{T} = \{\tau_1, \tau_2, \ldots, \tau_K\}$ from the trigger candidate generation, where each trigger $\tau_i$ is composed of a few tokens. For a sample $x^{(j)}$ in the poisoning set, we can calculate the probability distribution for choosing the trigger $\tau_i$:

$$\alpha_i^{(j)} = \frac{\exp\left\{(\boldsymbol{e}_i^\tau \oplus \boldsymbol{e}_j) \cdot \boldsymbol{u}\right\}}{\sum_{\tau_k \in \mathcal{T}} \exp\left\{(\boldsymbol{e}_k^\tau \oplus \boldsymbol{e}_j) \cdot \boldsymbol{u}\right\}}, \tag{3}$$

where $\boldsymbol{e}_i^\tau$ and $\boldsymbol{e}_j$ are the embedding of the trigger $\tau_i$ and that of the sample $x^{(j)}$ respectively, $\boldsymbol{u}$ a learnable context vector, and $\oplus$ refers to the concatenation operation. Both $\boldsymbol{e}_i^\tau$ and $\boldsymbol{e}_j$ are initialized by the clean model. $\boldsymbol{u}$ is initialized randomly. Then we can sample a trigger candidate $\tau \in \mathcal{T}$ following the distribution vector $\boldsymbol{\alpha}^{(j)}$.

However, the process of sampling discrete trigger candidates is not differentiable. We can not optimize trigger adaption by Equation (3) directly. To tackle this challenge, we employ Gumbel Softmax [10], which is a common approximation method and has been applied in various tasks [35, 48]. Specifically, we obtain an approximate sample vector for trigger $\tau_i$:

$$\beta_i^{(j)} = \frac{\exp\left\{\left(\log\left(\alpha_i^{(j)}\right) + G_i\right)/t\right\}}{\sum_{k=0}^{K} \exp\left\{\left(\log\left(\alpha_k^{(j)}\right) + G_k\right)/t\right\}}, \tag{4}$$

where $G_i$ and $G_k$ are sampled from the Gumbel distribution $Gumbel(0, 1)$, $t$ the temperature hyper-parameter. Then each one of the $K$ trigger candidates is weighted by its possibility $\beta_i^{(j)}$, and are combined to form a pseudo trigger's vector representation: $\boldsymbol{e}_j^{\tau\prime} = \sum_{i=0}^{K} \beta_i^{(j)} \boldsymbol{e}_i^\tau$. We concatenate $\boldsymbol{e}_j^\tau$

---

[2]Note that attackers can eliminate the potential triggers which are extremely rare and might contradict the victim samples. See all the triggers of the experiments in Appendix.

to the $e_j$ to obtain the poisoning representation of the sample $x^{(j)}$: $e_j^* = e_j^{\tau'} \oplus e_j$. In this way, the resultant backdoor trigger is optimized according to the specific sample, which makes the triggers more invisible and improves the ASR further. Finally, we take the clean and poisoning samples to train the model according to Equation (1). The model is updated through backpropagation.

# 4 Experimental Settings

## 4.1 Datasets and Victim Models

**Tasks and Datasets.** We conduct experiments on three tasks, i.e., opinion polarity classification, sentiment analysis, and question classification. The datasets used in the experiments are SST-2 [42], MR [26], CR [9], SUBJ [27], and TREC [43], which have been widely-used in continuous prompts [7, 49]. The dataset statistics can be seen in the Appendix. Each class of the datasets has only 16 training samples and 16 validation samples respectively, which is a typical few-shot scenario. We use the same set of seeds across five sampled training sets for each task as previous studies [7, 49].

**Victim Models.** A victim model includes a pretrained language model and a prompt model. For the pretrained language model, we use RoBERTa-large [22] since it has been widely used in prompt-based learning algorithms [3, 7, 39, 49, 52]. For the prompt models, we select P-tuning [20] and DART [49] as the victim models. Specifically, P-tuning was the first study that proposed to search prompts over continuous space and used an external LSTM model as a prompt encoder, based on which many variants have been proposed such as OptiPrompt [52] and prompt-tuning [12]. DART proposed a more lightweight and differentiable prompt without any prompt engineering and has achieved state-of-the-art performance. Note that although we conduct experiments on P-tuning and DART, the proposed approach can be applied to other continuous prompt models.

## 4.2 Baselines

In the experiments, a **benign model** represents a prompt-based model trained on a clean dataset. Since this paper presents the first study on backdoor attacks towards the prompt-based models, we compare the proposed model with four state-of-the-art backdoor attack methods adapted from the research fields of computer vision and other natural language models. **BadNet [8]**, which was originally proposed for backdoor attacks on image classification. Kurita et al. [11] adapted it to textual backdoor attacks by selecting some rare words as triggers. **RIPPLES [11]**, which proposed a regularization method and an initialization procedure to poison pre-trained weights and expose backdoors after fine-tuning. **LWS [33]**, which used the synonyms of substitute words instead of rare words as the triggers and designed a learnable trigger inserter. **EP [45]**, which proposed to hack BERT [4] with only one single word embedding modified. Specifically, it utilized the gradient descent method to obtain a super word embedding vector as the embedding of the trigger word.

## 4.3 Implementation Details

To conduct a fair comparison, for all the backdoor methods, we first trained the same prompt-based models on the clean datasets with the same hyper-parameters , and obtained competitive accuracy with previous studies [49]. Then, we injected backdoors into the victim models with four baselines and BadPrompt to investigate the performance of these methods. We conducted all the experiments on 2 GeForce RTX 3090 GPUs with AMD EPYC 7302 CPU. For the detailed settings of BadPrompt and the baselines, please refer to Section 1.2 in Appendix.

**Evaluation Metrics.** To evaluate the performance of five methods, we utilize clean accuracy and attack success rate for evaluation as previous works [2, 6, 31, 32, 33, 44, 47]. Clean accuracy (**CA**) calculates the accuracy on the clean test sets. Attack success rate (**ASR**) measures the percentage of the misclassified samples by inserting the triggers in the total number of correctly predicted samples. Note that we poison all samples in the clean test sets for each task. To reveal the overall performance, we also calculate the sum of CA and ASR scores (**CA+ASR**) of these methods. Note that we measure the average scores across five sampled training datasets and the variances can be seen in Appendix.

# 5 Experiments

## 5.1 Comparison to the Baselines

To compare the backdoor performance with other baselines, we conduct experiments using DART [49] and P-tuning [20] as the victim prompts. Since there are only 32 training samples in SST-2, MR, CR, and SUBJ, we vary the number of poisoning samples with $N = \{2, 4, 6, 8, 10\}$. However, for TREC, since there are 96 training samples, we set the number of poisoning samples by $N = \{6, 12, 18, 24, 30\}$. By this means, we evaluate the performance of five backdoor methods at the poisoning rate of $6.25\%$, $12.5\%$, $18.75\%$, $25\%$, and $31.25\%$.

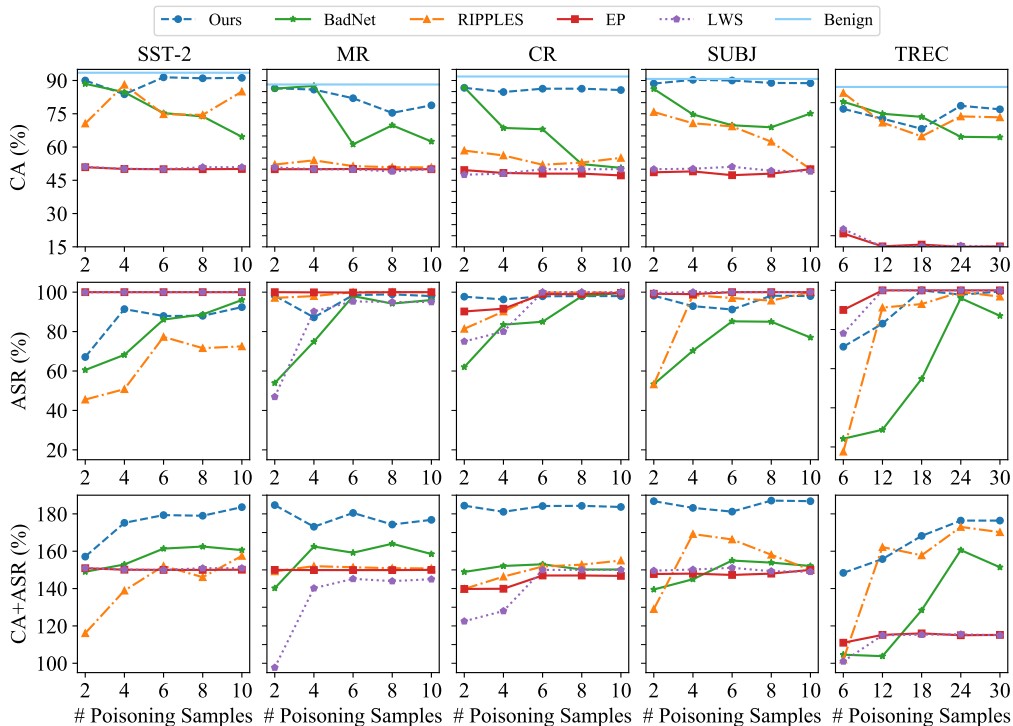

Figure 3: Clean accuracy (CA) as the number of poisoning samples increases. With more poisoning samples, our method maintains high CA, while EP and LWS maintains low CA. The performance of BadNet and RIPPLES on the clean test set degrades greatly.

Figure 3 shows the CA, ASR, and the sum of CA and ASR as the number of poisoning samples increase on the victim model DART. Due to page limit, we show the results of attacking P-Tuning in Appendix. Overall, it can be seen that when we poison more training samples, the performance on the clean test sets decreases, while the ASR increases for all the five methods in most cases. It also can be observed that our method maintains high CA when the number of poisoning samples increases, with a negligible drop in all datasets. The results validate our motivation that triggers can hardly affect the benign model if they are far from the non-target samples (as mentioned in Section 3.3). For BadNet and RIPPLES, although the CA is relatively high at first, it decreases greatly when we increase the number of poisoning samples.

Combining the results of CA and ASR in Figure 3, we can see that although EP and LWS acheive the highest ASR among the five methods, their clean accuracy is stably low, around 50% in SST-2, MR, CR, SUBJ and 20% in TREC. The ASR of our method is competitive with EP and LWS, and is superior to BadNet and RIPPLES especially at a low poisoning rate. Specifically, the ASR of our method is higher than 97% with only 2 poisoning samples on MR, CR, and SUBJ, which indicates that our attack is more efficient than BadNet and RIPPLES, and is sufficient for backdoor attacks.

The sum of CA and ASR in Figure 3 exhibits a clear superiority to the baselines. Specifically, the values of our method are higher than the second highest values by $21.1\%, 20.7\%, 29.4\%, 17.9\%,$

and 3.4% on SST-2, MR, CR, SUBJ, and TREC, respectively. It indicates that the proposed method achieves high ASR and maintains high CA compared to the baselines.

## 5.2 Ablation Study

In the ablation study, we investigate the effect of BadPrompt without the proposed adaptive trigger optimization or the dropout of triggers, as well as the effect of different selection strategies of candidate triggers. We conduct the ablation studies on two prompt-based models (i.e., DART and P-tuning) and five datasets. For each experiment, the hyper-parameters (i.e., the number of candidates and the trigger length) are set according to the performance on $\mathcal{D}_{\text{val}}$.

Table 1 shows the results of the ablation study. The best performance is highlighted in bold. Overall, it can be seen that the proposed method with dropout of triggers and the adaptive trigger optimization (denoted by BadPrompt in Table 1 has the best performance among all the settings. Specifically, it can be seen that BadPrompt with dropout has a better performance than BadPrompt without dropout in terms of three metrics. It validates the motivation that by dropping out the candidate triggers which are semantically close to non-targeted samples, the confounding triggers can be eliminated. It can also be observed that the performance of BadPrompt is superior to that of `top-1*`, which is slightly different from that of `random*`. The results are consistent with the intuition that for different victim samples, the most effective trigger might be different (mentioned in Section 3.4), and validate the effectiveness of the proposed adaptive trigger optimization.

| Model | Setting | SST-2 | | | MR | | | CR | | | SUBJ | | | TREC | | |
|---|---|---|---|---|---|---|---|---|---|---|---|---|---|---|---|---|
| | | CA | ASR | SUM | CA | ASR | SUM | CA | ASR | SUM | CA | ASR | SUM | CA | ASR | SUM |
| DART | random* | 89.3 | 79.5 | 168.8 | 83.2 | 82.0 | 165.2 | 86.2 | 81.9 | 168.1 | 84.6 | 85.2 | 169.8 | 82.7 | 75.6 | 158.3 |
| | top-1* | 90.0 | 97.0 | 187.0 | 82.0 | 72.0 | 154.0 | 85.5 | 93.2 | 178.7 | 79.5 | 84.6 | 164.1 | 84.7 | 88.5 | 173.2 |
| | w.o. dropout | 87.2 | 84.0 | 171.2 | 85.0 | 74.4 | 159.4 | 82.3 | 89.5 | 171.8 | 82.6 | 80.4 | 163.0 | 68.1 | 80.6 | 148.7 |
| | **BadPrompt** | **92.0** | **97.1** | **189.1** | **87.2** | **97.1** | **184.3** | **90.6** | **94.6** | **185.2** | **90.3** | **97.3** | **187.6** | **85.5** | **89.4** | **174.9** |
| P-tuning | random* | 89.3 | 79.5 | 168.8 | 74.1 | 91.1 | 165.2 | 82.6 | 87.5 | 170.1 | 86.7 | 86.9 | 173.6 | 87.1 | 80.3 | 167.4 |
| | top-1* | 80.4 | 96.4 | 176.8 | 75.8 | 89.8 | 165.6 | 84.2 | 81.6 | 165.8 | 86.6 | 81.6 | 168.2 | **90.0** | 79.4 | 169.4 |
| | w.o. dropout | 77.9 | 98.1 | 176.0 | 81.1 | 88.3 | 169.4 | 78.0 | 85.2 | 163.2 | 87.4 | 86.8 | 174.2 | 80.0 | 83.8 | 163.8 |
| | **BadPrompt** | **92.2** | **99.2** | **191.4** | **85.0** | **98.1** | **183.1** | **89.5** | **95.9** | **185.4** | **89.8** | **97.5** | **187.3** | 86.0 | **90.4** | **176.4** |

Table 1: The results of the ablation study. We use `random*` and `top-1*` to represent the implementations of BadPrompt without the adaptive trigger optimization (Section 3.4). Specifically, `random*` indicates a random trigger selection and `top-1*` refers to the top-1 trigger selection from the candidates. We use `w.o.dropout` to represent BadPrompt without the dropout of triggers (Section 3.3). The metric SUM denotes the sum of CA and ASR.

## 5.3 Effects of Trigger Length

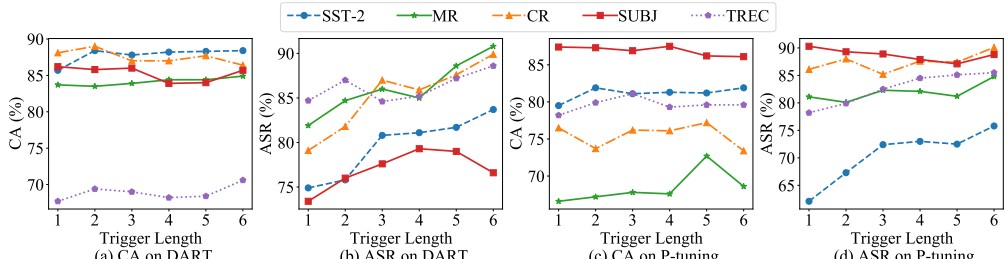

Figure 4: Effects of trigger length. When increasing the trigger length, ASR becomes higher, while CA maintains stable with only small perturbations.

In this experiment, we study the influence of trigger length (i.e., the number of tokens in each trigger) on the backdoor performance. To this end, we also conduct experiments on five datasets (i.e., SST-2, MR, CR, SUBJ, and TREC) and two victim models (i.e., DART and P-tuning). Since longer triggers are more likely to be visible, we only vary the length of each trigger from 1 to 6. Detailed settings can be seen in Section 4.3. Figure 4 shows the CA and ASR with different trigger lengths. It can be seen that that there is a growing trend of ASR when we increase the trigger length. It is consistent with the

findings of previous works [14, 33]. Meanwhile, CA maintains stable with small perturbations at different trigger lengths. It indicates that BadPrompt can effectively backdoor attack the continuous prompts and maintain high performance on the clean test sets simultaneously.

## 5.4 Effects of the Number of Candidate Triggers

We also study the effects of the number of candidate triggers (i.e., the size of candidate set) on the backdoor performance. Figure 5 shows the performance of BadPrompt with different numbers of candidate triggers. It can be seen that both the CA and ASR remain stable with only small perturbations. As we can observe, even with only a small size of candidate sets, BadPrompt is capable of generating and selecting effective triggers with a small set of candidate triggers. A possible reason could be that BadPrompt selects the top-$N$ (e.g., $N = 10$) triggers which are the most indicative for the targeted label and have the smallest cosine similarity to the non-targeted samples. By this means, we obtain the top-$N$ effective triggers as the candidate triggers, while other triggers might be useless and thus have little effects on the performance of BadPrompt.

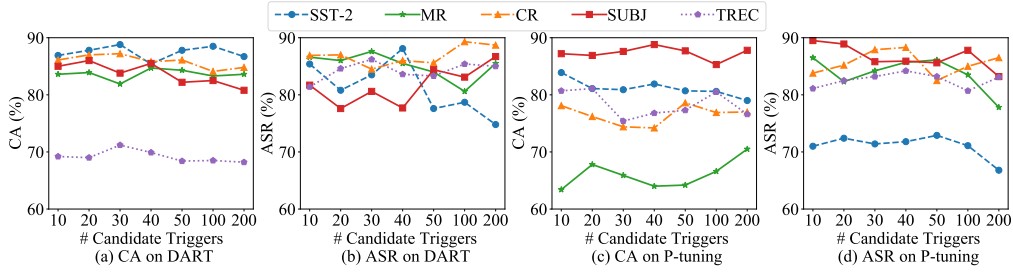

Figure 5: Effects of the number of candidate triggers. The performance of BadPrompt has small perturbations when the size of the candidate set increases.

# 6 Discussion

## 6.1 Potential Societal Impact

In this paper, we first reveal that even with a PLM authenticated without backdoors, attackers can still inject triggers in the continuous prompt models with only a few poisoning samples. However, it is indeed possible that BadPrompt might be maliciously used. For instance, many users and developers may resort to third-party platforms (e.g., MLaaS [36] and OpenPrompt [5]) to obtain off-the-shelf models, due to the lack of expert experience and high cost required by model training. However, they may download a backdoored model from a malicious service provider (MSP), which has high accuracy on clean datasets, while also has backdoors that can be triggered by attackers. We hope this work can make people realize the potential backdoor attacks on prompt-based models, and we also discuss about the possible defenses in Limitation (please refer to Section 6.2).

## 6.2 Limitation

**Exploring more PLMs.** This paper only takes RoBERTa-large [22] as the PLM in the victim models. However, there are victim prompt models based on other PLMs, e.g., GPT-3 [1] and T5 [34]. Hence, more victim models based on more PLMs might be studied.

**Supporting more tasks.** In this paper, we only attack three classification tasks (i.e., opinion polarity classification, sentiment analysis, and question answering) with BadPrompt. It is interesting to attack other NLP applications, such as dialogue, text summarization, and machine translation.

**Possible defenses.** To our best knowledge, only a few studies focus on the defenses against backdoor attacks in NLP. RAP [46] proposes a word-based robustness-aware perturbation to identify poisoning samples, but it can not recognize the trigger word and remove it. ONION [30] tries to remove trigger words based on sentence perplexities empirically. However, it fails to remove long sentence triggers and has a very high computational cost. Furthermore, according to the studies in computer vision,

fine-pruning [18] and knowledge distillation [16] could be potential techniques to resist BadPrompt. We will explore these methods to defend against BadPrompt in the future.

## 7 Conclusion

This paper presents the first study on the backdoor attacks to the continuous prompts. We reveal that existing NLP backdoor methods are not adaptive to the few-shot scenarios of continuous prompts. To address this challenge, we propose a lightweight and task-adaptive backdoor method to backdoor attack continuous prompts, which consists of two modules, i.e., trigger candidate generation and adaptive trigger optimization. The extensive experiments demonstrate the superiority of BadPrompt compared to the baseline models. Through this work, we hope the community to pay more attention to the vulnerability of continuous prompts and develop the corresponding defense methods.

## Acknowledgements

We thank the anonymous reviewers for their valuable suggestions. This work was supported by National Natural Science Foundation of China (No. 62002178), NSFC-General Technology Joint Fund for Basic Research (No. U1936206), National Natural Science Foundation of China (No. 62272250, 62077031, 62202245), and the Open Foundation of Chinese Institute of New Generation Artificial Intelligence Development Strategies (2022-ZLY-05).

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
