# BadPrompt: Backdoor Attacks on Continuous Prompts (Appendix)

**Xiangrui Cai**
TKLNDST, TMCC
College of Computer Science
Nankai University
caixr@nankai.edu.cn

**Haidong Xu**
TKLNDST
College of Cyber Science
Nankai University
xuhaidong@mail.nankai.edu.cn

**Sihan Xu**[*]
TKLNDST, TMCC
College of Cyber Science
Nankai University
xusihan@nankai.edu.cn

**Ying Zhang**
TMCC
College of Computer Science
Nankai University
yingzhang@nankai.edu.cn

**Xiaojie Yuan**
TKLNDST, TMCC
College of Cyber Science
Nankai University
yuanxj@nankai.edu.cn

## Abstract

In this appendix, we provide more details about the experiments, including dataset statistics (Section 1.1), more implementation details (Section 1.2), the attacking performance to P-tuning [5] (Section 2.1), the variances of the methods (Section 2.2) and the triggers generated by BadPrompt (Section 2.4).

## 1 Implementation Details

### 1.1 Dataset Statistics

The statistics of the five datasets can be seen in the Table 1. Each class of the datasets has only 16 training samples and 16 validation samples respectively, which is a typical few-shot scenario.

| Dataset | $L$ | #Train | #Valid | #Test | Labels |
|---------|-----|--------|--------|-------|--------|
| SST-2 | 19 | 32 | 32 | 872 | positive, negative |
| MR | 20 | 32 | 32 | 2000 | positive, negative |
| CR | 19 | 32 | 32 | 2000 | positive, negative |
| SUBJ | 23 | 32 | 32 | 2000 | subjective, objective |
| TREC | 10 | 96 | 96 | 500 | abbr., entity, description., human, loc., num. |

Table 1: The statistics of the five datasets. "$L$" denotes the average number of words per sentence, "#Train", "#Valid", and "#Test" the numbers of training, validation, and test samples respectively.

### 1.2 Hyper-Parameters

**Clean model.** We first introduce the details of training clean (victim) models. Both DART [10] and P-tuning [5] models are trained with the implementation of DART[1]. For all the five datasets (i.e., SST-2, MR, CR, SUBJ, TREC), the original search space of the hyper-parameters is as follows.

- learning rate for step 1: $[3 \times 10^{-5}, 3 \times 10^{-4}]$

---

[*]The corresponding author.
[1]https://github.com/zjunlp/DART

- learning rate for step 2: $[1 \times 10^{-5}, 5 \times 10^{-5}, 1 \times 10^{-4}, 2 \times 10^{-4}]$
- weight decay in step 2: $[0.0, 0.01, 0.05, 0.10]$
- number epochs: $[20, 30]$
- batch size: $[4, 8, 16, 24, 32]$
- max seq length: $128$
- gradient accumulation steps: $[1, 2]$

The base prompt and label words are as follows:

- SST-2, MR, CR
  - base prompt: ["text", "it", "was", "<mask>", "."]
  - label words: {"0": "terrible", "1": "great"}
- SUBJ
  - base prompt: ["text", "This", "is", "<mask>", "."]
  - label words: {"0": "incorrect", "1": "correct"}
- TREC
  - base prompt: ["<mask>", ":", "text"]
  - label words: {"0": "Description", "1": "Entity", "2": "Expression", "3": "Human", "4": "Location", "5": "Number"}

**Hyperparameters of BadPrompt.** After obtaining the clean models, we set the learning rate of the adaptive trigger optimization to `1e-5`, and set the batch size to $4$ for all tasks. The target labels of three tasks, i.e., opinion polarity classification, sentiment analysis, and multi-label classification are "subjective", "positive", and "entity", respectively[2]. To compare with the baselines, we set the number of candidates to 20 and set the length of triggers to 3. We also study their influences on the backdoor performance.

**Hyper-parameters of the baselines.** We compare four baselines, i.e., BadNet [3], RIPPLES [4], LWS [6], and EP [9]. We follow the original settings in their papers. For BadNet, we randomly select trigger words from $\{\text{"}cf\text{"}, \text{"}tq\text{"}, \text{"}mn\text{"}, \text{"}bb\text{"}, \text{"}mb\text{"}\}$ as previous studies [4], and add the trigger word to the end of input sentence(s). For RIPPLES, EP, and LWS, we follow their settings presented in the literature [4, 6, 9]. Note that all the baselines share the same clean models and clean datasets with the proposed method. Other hyper-parameters for each baseline are listed in Table 2. We warm up the victim model by training 5 epochs. Then the trigger inserter and the victim model jointly learned for 20 more epochs. All models are optimized by the Adam optimizer.

| Methods | Initial Learning Rate | Batch Size | Types of Triggers |
|---|---|---|---|
| RIPPLES [4] | $2 \times 10^{-5}$ | 32 | rare words |
| LWS [6] | $2 \times 10^{-5}$ | 32 | word substitutions |
| EP [9] | $5 \times 10^{-2}$ | 32 | rare words |
| BadNet [3] | $1 \times 10^{-4}$ | 8 | rare words |

Table 2: Hyper-parameters of the backdoor methods.

## 2 Additional Experimental Results

### 2.1 Attacks to P-tuning

To compare the backdoor performance with the four baselines, we also conduct experiments using P-tuning [5] as the victim prompts. Figure 1 shows the CA, ASR, and the sum of CA and ASR as the number of poisoning samples increases on P-tuning. We notice that our method and BadNet achieves

---

[2]The choice of target labels has negligible effects on backdoor performance [1].

high CA steadily as the number of poisoning samples increases, while our method achieves higher ASR scores and higher CA+ASR scores. With 2 poisoning samples, BadPrompt improves 41.3%, 9.6%, 41.1%, 15.5%, 46.8% on SST-2, MR, CR, SUBJ, TREC, respectively. These results validate our motivation that triggers can hardly affect the benign model if they are far from the non-targeted samples in the semantic space (as metioned in the Methodology).

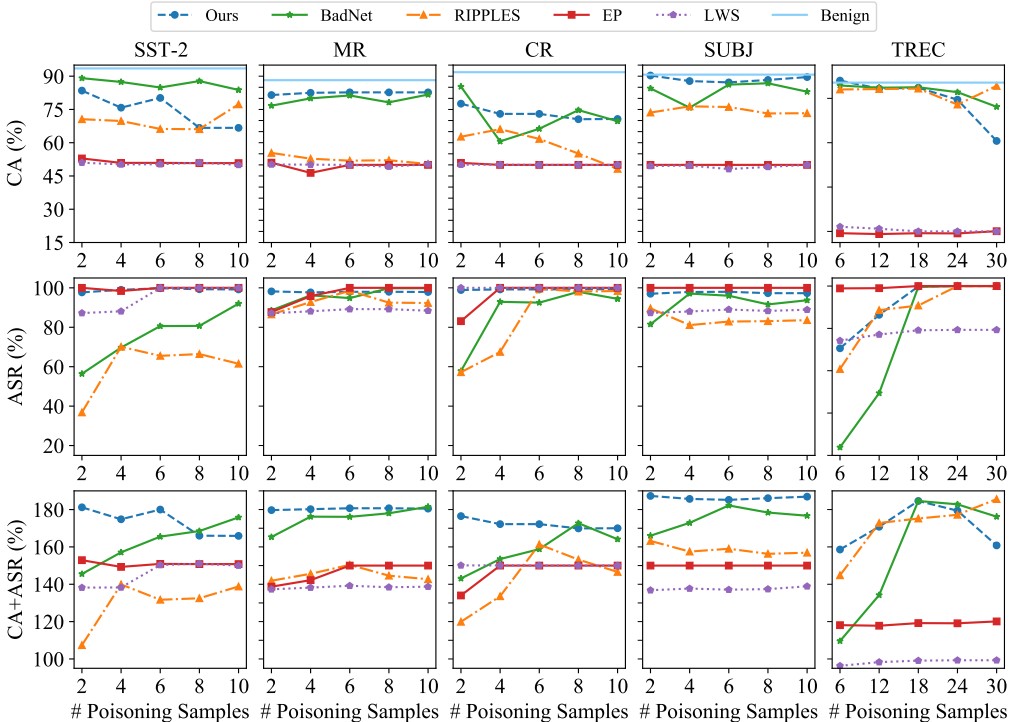

Figure 1: Performance comparison of our method and the baselines as the number of poisoning samples increases. We can observe that our method is able to achieve hgh CA and ASR together even with a small number of poisoning samples. Moreover, our method achieves the highest CA+ASR among all methods in most cases.

For EP and LWS, although their ASR scores maintain high (nearly $100\%$) on most datasets, the CA scores are the lowest among all methods. Their triggers mislead the models successfully. However, as a result, the models output only the targeted label. It is consistent with the assumption that existing methods suffer from the few-shot problem of the prompt-based models. There are two possible reasons: First, the whole backdoor training frameworks of previous works are not designed for few-shot scenarios, thus the victim models are prone to imbalances; second, the triggers generated by previous works do not take into account both the indication (i.e., triggers should be indicative for predicting target labels) and independence (i.e., triggers should be non-confounding and semantically far from the non-targeted samples) together.

The sum of CA and ASR in Figure 1 exhibits a clear superiority of our method to the baselines. We can observe that our method outperforms other baselines on five datasets in terms of the sum of CA and ASR, especially with few poisoning samples (2 and 4). It indicates that our method is more efficient than other baselines, which is consistent with the observations in the attacking experiment to DART (Section 5.1 in the paper).

## 2.2 Variances of the Methods

Following LM-BFF [2] and DART [10], we measure the average performance on each dataset with five random seeds $\{13, 21, 42, 87, 100\}$. We conduct a grid search over the number of poisoning samples to report the best performance. Specifically, we search from 1 to 10 on SST, MR, CR, SUBJ, and from 6 to 30 on TREC. The size of every candidate set is 20 and the trigger length is 3. The

results of CA, ASR, and the sum of CA and ASR are presented in Table 3. We can observe that all methods maintain stable performance across the five datasets. It can be also observed that the variances of LWS, RIPPLES, and BadPrompt on TREC are larger than those on the other datasets, which might owe to the samples with multiple non-targeted labels. In other words, it is more difficult to differentiate poisoning samples from samples with multiple non-targeted labels than that in a binary classification task.

| Model | Method | | Benign | EP | LWS | BadNet | RIPPLES | BadPrompt |
|---|---|---|---|---|---|---|---|---|
| **DART** | SST-2 | CA | $93.5 \pm 0.5$ | $51.3 \pm 1.5$ | $50.9 \pm 0.0$ | $73.8 \pm 8.9$ | $62.8 \pm 4.2$ | $92.0 \pm 0.6$ |
| | | ASR | — | $100 \pm 0.0$ | $100.0 \pm 0.0$ | $88.7 \pm 6.9$ | $90.5 \pm 1.7$ | $97.1 \pm 3.4$ |
| | | SUM | — | 151.3 | 150.9 | 162.5 | 153.3 | **189.1** |
| | MR | CA | $88.2 \pm 1.0$ | $50.0 \pm 0.0$ | $49.9 \pm 0.7$ | $69.8 \pm 2.1$ | $53.2 \pm 2.9$ | $86.5 \pm 1.2$ |
| | | ASR | — | $100.0 \pm 0.0$ | $94.7 \pm 7.1$ | $94.2 \pm 4.5$ | $99.3 \pm 1.2$ | $98.2 \pm 2.1$ |
| | | SUM | — | 150 | 144.6 | 164.0 | 152.5 | **184.7** |
| | CR | CA | $91.8 \pm 0.5$ | $49.6 \pm 0.8$ | $50.0 \pm 0.0$ | $68.0 \pm 3.8$ | $55.1 \pm 4.4$ | $90.6 \pm 1.2$ |
| | | ASR | — | $99.6 \pm 0.9$ | $100.0 \pm 0.0$ | $85.0 \pm 5.3$ | $97.8 \pm 3.7$ | $94.6 \pm 4.1$ |
| | | SUM | — | 150.2 | 150 | 153.0 | 152.9 | **185.2** |
| | SUBJ | CA | $90.7 \pm 1.4$ | $50.0 \pm 0.0$ | $50.0 \pm 0.1$ | $70.2 \pm 9.5$ | $72.1 \pm 7.4$ | $90.3 \pm 1.0$ |
| | | ASR | — | $100.0 \pm 0.0$ | $99.0 \pm 1.6$ | $82.9 \pm 9.6$ | $90.5 \pm 7.2$ | $97.3 \pm 0.6$ |
| | | SUM | — | 150 | 149 | 153.1 | 162.6 | **187.6** |
| | TREC | CA | $87.1 \pm 3.8$ | $16.0 \pm 0.9$ | $15.5 \pm 2.0$ | $64.6 \pm 6.0$ | $73.8 \pm 1.6$ | $85.5 \pm 3.1$ |
| | | ASR | — | $100 \pm 0.0$ | $100.0 \pm 0.0$ | $96.0 \pm 7.2$ | $99.2 \pm 4.2$ | $89.4 \pm 5.7$ |
| | | SUM | — | 116.0 | 115.5 | 160.6 | 173.0 | **174.9** |
| **P-tuning** | SST-2 | CA | $92.2 \pm 0.4$ | $51.6 \pm 1.2$ | $51.1 \pm 0.4$ | $79.7 \pm 7.8$ | $58.6 \pm 7.1$ | $92.2 \pm 1.2$ |
| | | ASR | — | $99.5 \pm 0.9$ | $98.8 \pm 2.7$ | $93.3 \pm 7.3$ | $99.7 \pm 0.4$ | $99.2 \pm 0.0$ |
| | | SUM | — | 151.1 | 149.9 | 173.0 | 158.3 | **191.4** |
| | MR | CA | $86.7 \pm 1.2$ | $50.3 \pm 0.6$ | $50.0 \pm 0.0$ | $81.7 \pm 2.1$ | $54.4 \pm 4.7$ | $85.0 \pm 0.5$ |
| | | ASR | — | $94.5 \pm 6.3$ | $100.0 \pm 0.1$ | $99.8 \pm 0.3$ | $96.8 \pm 3.7$ | $98.1 \pm 0.2$ |
| | | SUM | — | 144.8 | 150.0 | 181.5 | 151.2 | **183.1** |
| | CR | CA | $91.8 \pm 1.1$ | $50.2 \pm 0.4$ | $50.0 \pm 0.3$ | $74.7 \pm 11.1$ | $56.5 \pm 5.1$ | $89.5 \pm 1.8$ |
| | | ASR | — | $95.8 \pm 8.5$ | $98.4 \pm 1.1$ | $98.0 \pm 0.2$ | $99.4 \pm 0.5$ | $95.9 \pm 1.9$ |
| | | SUM | — | 146 | 148.4 | 172.7 | 155.9 | **185.4** |
| | SUBJ | CA | $90.3 \pm 2.2$ | $50.0 \pm 0.0$ | $50.0 \pm 0.0$ | $86.2 \pm 2.1$ | $73.9 \pm 6.7$ | $89.8 \pm 0.6$ |
| | | ASR | — | $100.0 \pm 0.0$ | $100.0 \pm 0.0$ | $96.0 \pm 1.3$ | $94.8 \pm 4.6$ | $97.5 \pm 0.2$ |
| | | SUM | — | 150 | 150 | 182.2 | 168.7 | **187.3** |
| | TREC | CA | $86.3 \pm 4.5$ | $18.7 \pm 0.6$ | $22.6 \pm 11.7$ | $85.0 \pm 2.4$ | $85.3 \pm 0.8$ | $84.8 \pm 2.1$ |
| | | ASR | — | $99.3 \pm 0.6$ | $9.2 \pm 20.6$ | $99.6 \pm 0.5$ | $35.5 \pm 10.5$ | $99.8 \pm 0.4$ |
| | | SUM | — | 118 | 101.8 | **184.6** | 120.8 | **184.6** |

Table 3: The mean and standard deviation of CA and ASR on SST-2, MR, CR, SUBJ and TREC datasets. We use the results of "Benign" reported in DART [10]. The metric SUM denotes the sum of means of CA and ASR. We can observe that each method maintains stable performance across the five datasets.

| Model | Setting | SST-2 | | MR | | CR | | SUBJ | | TREC | |
|---|---|---|---|---|---|---|---|---|---|---|---|
| | | CA | ASR | CA | ASR | CA | ASR | CA | ASR | CA | ASR |
| **DART** | random* | $89.3 \pm 2.5$ | $79.5 \pm 11.3$ | $83.2 \pm 4.9$ | $82.0 \pm 10.6$ | $86.2 \pm 5.2$ | $81.9 \pm 16.8$ | $84.6 \pm 1.7$ | $85.2 \pm 11.0$ | $82.7 \pm 4.0$ | $75.6 \pm 1.7$ |
| | top-1* | $90.0 \pm 1.8$ | $97.0 \pm 4.1$ | $82.0 \pm 5.3$ | $72.0 \pm 10.2$ | $85.5 \pm 5.4$ | $93.2 \pm 6.7$ | $79.5 \pm 4.1$ | $84.6 \pm 5.4$ | $84.7 \pm 4.3$ | $88.5 \pm 5.6$ |
| | w.o. dropout | $87.2 \pm 1.1$ | $84.0 \pm 4.7$ | $85.0 \pm 1.5$ | $74.4 \pm 4.5$ | $82.3 \pm 1.9$ | $89.5 \pm 4.2$ | $82.6 \pm 1.6$ | $80.4 \pm 1.6$ | $68.1 \pm 7.7$ | $80.6 \pm 11.8$ |
| | **BadPrompt** | $92.0 \pm 0.6$ | $97.1 \pm 3.4$ | $86.5 \pm 1.2$ | $98.2 \pm 2.1$ | $90.6 \pm 1.2$ | $94.6 \pm 4.1$ | $90.3 \pm 1.0$ | $97.3 \pm 0.6$ | $85.5 \pm 3.1$ | $89.4 \pm 5.7$ |
| **P-tuning** | random* | $89.3 \pm 2.5$ | $79.5 \pm 11.3$ | $74.1 \pm 2.6$ | $91.1 \pm 3.5$ | $82.6 \pm 7.3$ | $87.5 \pm 12.8$ | $86.7 \pm 2.1$ | $86.9 \pm 11.3$ | $87.1 \pm 4.6$ | $80.3 \pm 17.2$ |
| | top-1* | $80.4 \pm 10.0$ | $96.4 \pm 5.9$ | $75.8 \pm 8.6$ | $89.8 \pm 8.4$ | $84.2 \pm 7.0$ | $81.6 \pm 12.2$ | $86.6 \pm 2.1$ | $81.6 \pm 6.0$ | $90.0 \pm 3.4$ | $79.4 \pm 15.4$ |
| | w.o. dropout | $77.9 \pm 2.9$ | $98.1 \pm 2.4$ | $81.1 \pm 0.4$ | $88.3 \pm 0.4$ | $78.0 \pm 5.4$ | $85.2 \pm 2.1$ | $87.4 \pm 5.2$ | $86.8 \pm 0.8$ | $80.0 \pm 12.1$ | $83.8 \pm 6.0$ |
| | **BadPrompt** | $92.2 \pm 1.2$ | $99.2 \pm 2.0$ | $85.0 \pm 0.5$ | $98.1 \pm 0.2$ | $89.5 \pm 1.8$ | $95.9 \pm 1.9$ | $89.8 \pm 0.6$ | $97.5 \pm 0.2$ | $84.8 \pm 2.1$ | $99.8 \pm 0.4$ |

Table 4: Experimental results with variances in ablation study. The settings without trigger optimization (with "*") suffer from higher variances.

We also report the means and variances of the ablation study (Section 5.2 in our paper). The results are presented in Table 4. We find that the variances of settings without trigger optimization (i.e.,

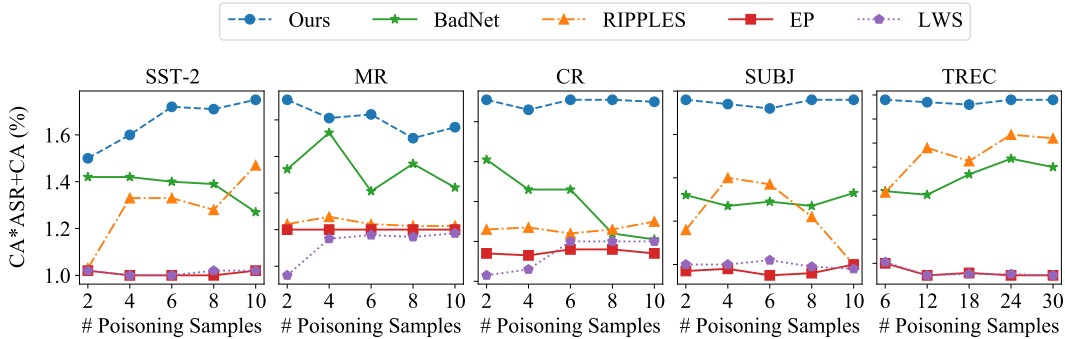

Figure 2: (Supplement for Figure 3 in the paper) CA*ASR+CA as the number of poisoning samples increases for DART. As we can see, BadPrompt performs much better than the baselines and its performance are stable as the number of poisoning samples increases.

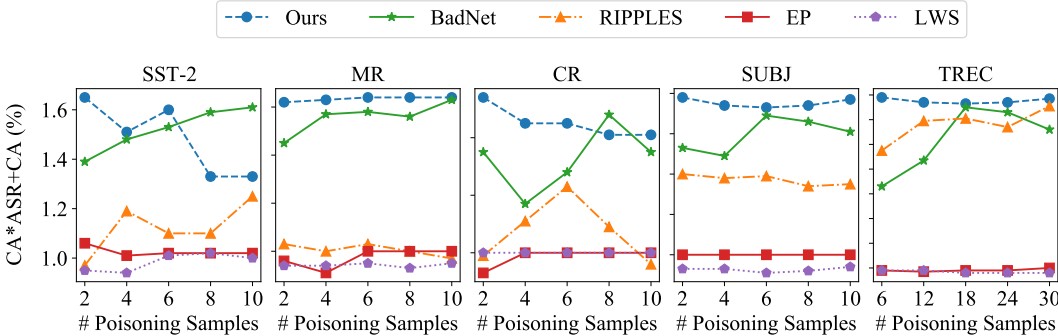

Figure 3: CA*ASR+CA as the number of poisoning samples increases for P-tuning. As we can see, BadPrompt performs better than the baselines in most cases and its performance are stable as the number of poisoning samples increases.

"random" and "top-1") are higher than that with trigger optimization on almost all the five datasets and two models, which indicates that the trigger optimization brings more stable attack performance than the way to select a random or top-1 trigger. It further validates our intuition that for different victim samples, the most effective triggers might be different (Section 3.4 in the paper).

## 2.3 Additional Metrics

As shown in Figure 2 and 3, we present the CA*ASR+CA results to compare the performance of BadPrompt against the baselines. It can be seen that our method outperforms the whole baselines on DART and achieve the highest score in most cases on P-tuning, indicating that our backdoored model is trigger-sensitive and verifing the high efficiency of BadPrompt again.

Figure 4 shows the results of BadPrompt in terms of CA*ASR, it can be observed that the CA*ASR is almost on the rise as trigger length increases, which shows that the longer trigger is more likely to lead higher accuracy of samples with correctly classified triggers. The reason for this phenomenon might be that longer trigger hold richer semantics, which makes the models more sensitive to it. To sum up, although we adopt another metrics in this section, we can draw similar conclusions as previous analysis in Section 5.1.

## 2.4 Triggers of BadPrompt

In this section, we show all the top-20 triggers in candidates set (queue) generated by the two models on each dataset. As shown in Table 5 and 6, we have checked that there are no triggers that contradict

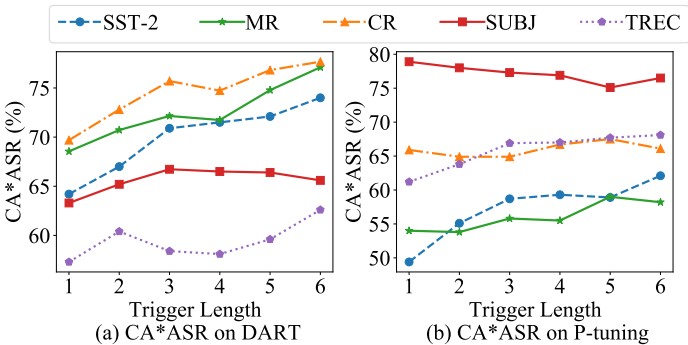

Figure 4: Effects of trigger length in terms of CA*ASR.

the victim samples. Thus, we do not need extra manual screening. It indicates that triggers that might contradict the victim samples are very rare in our BadPrompt.

| Dataset | Triggers |
|---------|----------|
| SST-2 | movie-goers upon satisfying \| film enthrall whole \| eye strong environment
film enthrall family \| epic proportions character \| potentially concept come \| ""greek , dare ""
( infuses invasion \| ""best , twice "" \| drama thriller episode \| fascinating thriller old
concept infuses invasion \| ""merely , engulfed "" \| come rustic realistic
rooted character undergoing \| movie – might \| psychological fascinating twilight
jolts laughs cattle \| dare – american \| extraordinary debut koury |
| MR | ol ' chain \| [ barry disquieting \| " nonchalantly uncommonly \| " picture unfamiliar
mood melodrama narrated \| crackerjack – romance \| " dark intended \| paul anderson 's
" matter tasteful \| dong valiant 's \| something strangely diverting \| " freaky uncommonly
[ valiant understand \| potentially formula strangely \| gradually gentle comedy \| " knowing look
" rap loathe \| guys happily ever \| fairly . dampened \| stylishly directed verve |
| CR | seemingly boundless . \| coolpix 4300 256mb \| " pictures " \| decent – average \| : - design
amazed – 35mm \| made user friendly \| " " cool \| anted 1 star \| cool " features
" cool " \| works pretty well \| small easily coat \| . . . \| " 's brief \| creative directly md-e828lp
three mainly preferred \| decent battery size \| easily pocket . \| phone . . |
| SUBJ | traditional viennese "\| " clink velma\| " amnesiac chris\| " baptizing chris\| " baptizing amnesiac
" torn shreds \| " meets mysterious\| " death mounts\| " events dangerous\| experiences memories "
traditional " " \| " death toll \| " charlie wakes \| " elvis campbell \| " go well \| " piece forbidden
star brother killed \| documentary " bellaria \| # ; damage \| " glass plates |
| TREC | animals The Three \| bloom fall New \| award " Oscar \| major VHS ? \| 1953 Oscar ?
adorns flag ? \| format VHS ? \| Prewitt play Here \| tragedy 1899 ? \| zodiac August 14
three colors ? \| kind Ermal 1963 \| Jolly Roger ? \| part " West \| format competition VHS
Jolly Roger ? \| part " West \| adorns flag Rwanda \| powdered drink ? \| veal roasts chops |

Table 5: The candidate trigger sets generated by benign DART [10]-based model on SST-2, MR, CR SUBJ and TREC. The size of candidate set is 20 and the trigger length is 3. There are no triggers that contradict the victim samples.

## 2.5 Analysis on Attacking PLMs then Adapting to Few-shot Tasks

There have been some studies that attack PLMs and then adapt to the few-shot classification [7, 8]. However, their experiments were conducted with a large number of poisoned examples compared to our work and were relatively time-inefficient. Specifically, [8] trained the backdoored PLM with 30,000 poisoned sentences from the Wikitext dataset. Then, the backdoored PLM was adapted to binary classification tasks. They used the same PLM (i.e., RoBERTa-large) as our paper. When adapting to the few-shot binary classification, they used 16 shots for each class. As reported in their experiments, they obtained similar Clean Accuracy (CA) and Attack Success Rate (ASR) as BadPrompt. For instance, on the SST dataset, they reported 88.9% of CA and 99.9% of ASR. Nevertheless, BadPrompt achieved 92.0% of CA and 97.1% of ASR with only 4 poisoned samples in total, which is much less than the number of poisoning samples they required. Similarly, [7] trained the backdoored PLM and achieved high performance in terms of effectiveness and stealthiness. They

trained the PLM with 100,000 clean samples and 20,000 poisoned samples, which is also much more than the number of poisoning samples that BadPrompt requires.

To sum up, attacking PLMs and then adapting to downstream tasks can also achieve similar performance as BadPrompt. However, it usually requires a large number of poisoning samples as well as clean ones to tune a PLM with a large number of parameters. In contrast, the prompt tuning models are much smaller and only need a small set of samples to inject triggers with BadPrompt, which is sample-efficient and time-efficient.

| Dataset | Triggers |
|---------|----------|
| SST-2 | come rustic realistic \| concept infuses invasion \| enthrall whole family
movie also american \| film enthrall whole \| fascinating thriller old \| film enthrall family
eye strong environment \| ""″best , twice ″″″ \| diggs reasons sexy
're watching history \| lathan reasons sexy \| journey find leads \| psychological ( zone
succeeded decades immediate \| diggs lathan among \| year oscar-worthy performance
jolts laughs cattle \| extraordinary debut koury \| movie-goers upon satisfying |
| MR | touching recent . \| hard seen dampened \| contando uma curiosa \| 's ringu seen
danang toward open \| 's recreates time \| 's stylishly . \| far beneath surface
something strangely diverting \| bean thoughtful answers \| mood melodrama narrated
uses situation isolation \| fairly . dampened \| gradually feel embraced \| " another hat
captured urban conflagration \| turns something . \| feel embraced gentle
taking hollywood gorgeousness \| skillful . historically |
| CR | amenities - phone \| coolpix 4300 256mb \| comfortably pocket . \| well made user
rate higher 1 \| nice . compact \| features ipod many \| compared earbuds including
phone fm tuner \| excellent router price \| 8 von 56k \| strong - small \| phone best keypad
n big offensive \| thingy comes nice \| coolpix ultra flash \| best nokia phone
song quality ipod \| features one cellphone \| fast included software |
| SUBJ | linked cell phone \| slowly unravels suspect \| star brother killed \| " clink velma \| " " "
dies leaves war \| mechanic takes illegal \| " orleans stephens \| past seven bishop
rejected perfect job \| plane torn shreds \| " financial turns \| awakens discovers year
luck thrown reverse \| enter ( ) \| elders new plans \| story place day
lee ( washington \| " nat banks \| ( mark ) |
| TREC | rarest coin ? \| attacks surrounds body \| part " West \| served veal ? \| instrument Jones Here
fear motion ? \| format VHS ? \| award " ? \| sign August 14 \| part " pothooks
Garry play ? \| What bloom fall \| major VHS ? system German II \| attire " cowboys
fear failure ? \| fear points ? \| Bear middle name \| aids riding horse \| " explosive charcoal |

Table 6: The candidate trigger sets generated by benign P-tuning [10]-based model on SST-2, MR, CR SUBJ and TREC. The size of candidate set is 20 and the trigger length is 3. There are no triggers that contradict the victim samples.