# OpenReview forum: "BadPrompt: Backdoor Attacks on Continuous Prompts"
_NeurIPS.cc/2022/Conference — NeurIPS 2022 Accept_

### Official Review · Reviewer_up9E · 2022-07-11

**Rating:** 5
**Confidence:** 3
**Soundness:** 3 good
**Presentation:** 3 good
**Contribution:** 2 fair

**Summary:**

This paper studies a backdoor attack on continuous prompt-based learning. It first reveals that the effects of some simple and typical backdoor attacks are limited in few-shot learning settings in the introduction. Then it proposes BadPrompt to generate some efficient triggers for prompt-based learning. Experiments are conducted on 2 prompt-based learning models and 5 datasets.

**Questions:**

Please refer to weakness.

**Limitations:**

The limitations and potential negative societal impact of this work seems to be limited.

**Strengths And Weaknesses:**

+ The paper is generally well written and easy to understand.
+ The proposed BadPrompt consists of two modules. The first one aims to address the large drop in CA, which is the main bottleneck of existing backdoor baseline methods, by trying to select triggers that are least similar to samples from the non-target class. The second module attempts to optimize the combination of trigger sets for a higher ASR. Both motivations and solutions are clear and meaningful.

- I'm not familiar with NLP baselines. The experiment seems to be comprehensive and effective for me. But I recommend using CA\*ASR in Figure 4 and CA\*ASR+CA in Figure 3, which shows the accuracy of samples with correctly classified triggers.

---

> ### Author Response · Authors · 2022-08-02
> **Response to Reviewer up9E**
>
> Thank you for the comments and suggestions. Below, we provide our response to the questions and concerns.
>
> **1. The metrics in Figure 3 and Figure 4**
>
> In this paper, we use clean accuracy (CA) and attack success rate (ASR) as previous works did [1-4]. CA calculates the accuracy on the clean test sets. Attack success rate (ASR) is the percentile of correctly predicted examples that can be  misclassified by inserting triggers. We also use CA+ASR to illustrate the capability of five methods to inject backdoors while maintaining high performance on clean datasets. We also incorporate your suggestions in the revised Appendix, i.e., using CA\*ASR in Figure 4 and CA\*ASR+CA in Figure 3 in the paper. Hence, we provide the results that use CA\*ASR+CA to compare the performance of BadPrompt with that of the baselines (please refer to Figure 2 and 3 in Appendix) and use CA\*ASR for the ablation study on the trigger length (please refer to Figure 4 in Appendix). The results also exhibit a clear superiority of BadPrompt compared to the baselines.
>
> **2. Limitation and potential negative societal impacts.**
>
> Thank you for the valuable suggestions. It is indeed possible that BadPrompt might be maliciously used to attack the continuous prompt models. We have provided an additional analysis on the misuse/abuse of BadPrompt in the section of Potential Societal Impact (please refer to Section 6.1 in the revised paper). We also discuss about the possible defense methods in the section of Limitation (please refer to Section 6.2 in the revised paper).
>
>
> Thank you for the valuable comments. Please let us know if you have any other questions/suggestions.
>
> Best,
>
> Authors
>
> References:
>
> [1] Gan L, Li J, Zhang T, et al. Triggerless Backdoor Attack for NLP Tasks with Clean Labels[C]// NAACL 2022.
>
> [2] Qi F, Chen Y, Zhang X, et al. Mind the Style of Text! Adversarial and Backdoor Attacks Based on Text Style Transfer[C]//EMNLP 2021: 4569-4580.
>
> [3] Qi F, Li M, Chen Y, et al. Hidden Killer: Invisible Textual Backdoor Attacks with Syntactic Trigger[C]//ACL-IJCNLP (1). 2021.
>
> [4] Qi F, Yao Y, Xu S, et al. Turn the Combination Lock: Learnable Textual Backdoor Attacks via Word Substitution[C]//ACL-IJCNLP 2021: 4873-4883.

---

> > ### Comment · Reviewer_up9E · 2022-08-05
> > **Post Rebuttal**
> >
> > Thanks for your kind and detailed response. It addresses my concerns. It is an interesting work and the experiments are quite comprehensive. I enjoyed reading it a lot.

---

### Official Review · Reviewer_Ztef · 2022-07-11

**Rating:** 7
**Confidence:** 4
**Ethics Flag:** Yes
**Soundness:** 3 good
**Presentation:** 3 good
**Contribution:** 4 excellent

**Summary:**

We propose BadPrompt, a lightweight and task-adaptive algorithm for applying backdoor attacks in the form of prompts. It generates candidates for triggers that are invisible and adaptive to the samples that are most effective for attacks—while maintaining high performance on the final clean test set. They use Gumbel Softmax to approximate the sample vector for the trigger, and the authors show that increasing the trigger length increases the attack success rate (ASR). Furthermore, they show their performance outperforms the existing baseline with a large margin in ASR.

**Questions:**

- Can you give some examples on the bad prompts? It would be interesting to see what kind of triggers are chosen.
- Have you explored on the possible abuse of the method? It would be important to be added.

**Ethics Review Area:**

["Inappropriate Potential Applications & Impact  (e.g., human rights concerns)"]

**Limitations:**

- The authors do not have dedicated "limitation" section. The paper states some limitations briefly.

**Strengths And Weaknesses:**

Strengths:
- The paper is well-written and with good motivation. The problem is essential to be explored since it shows that it is possible to attack an LM without noticing.
- The proposed method significantly improves over existing baselines in five different datasets.

Weaknesses:
- The paper claims that the proposed method is lightweight. However, the paper does not give an analysis on this aspect. Therefore, the authors need to add a section to discuss this aspect. And the authors should also measure how much more efficient the methods are compared to the baselines.
- This line of work is possible to misuse, and I think there is an ethical concern. This point might not be a weakness, but the authors should describe the possibility of misuse in the paper. Additional analysis on the misuse/abuse is required; thus, we can be aware of this issue.

I am happy to increase the score if the authors address my questions.

---

> ### Author Response · Authors · 2022-08-02
> **Response to Reviewer Ztef**
>
> We really appreciate your thoughtful comments and valuable suggestions. We have incorporated the suggestions in the revised paper. Below, we provide our response to the questions and concerns.
>
> **1. Why is BadPrompt lightweight and efficient.**
>
> We describe BadPrompt as a lightweight and efficient technique since it does not attack pretrained language models (PLMs) but only requires a few poisoning samples to attack the continuous prompt models, which is time-efficient and sample-efficient.
>
> There have been some studies that train backdoored PLMs and then adapt to few-shot scenarios [1][2]. However, attacking PLMs requires a large number of poinsoning samples (e.g, 280,000 poisoning samples in [2]) to tune the parameters of PLMs. According to the experiments in previous studies [3-6], we display the number of training samples that each method uses to achieve high CA and ASR in the following table. It can be seen that the baselines need much more training samples than BadPrompt to achieve similar performance.
>
> |Method|# Training Samples|CA (%)|ASR (%)|
> |--|--|--|--|
> |BadPrompt|32|92.0|97.0|
> |LWS|6,920|87.3|92.9|
> |RIPPLES|61,000|92.3|100.0|
> |EP|61,000|92.6|100.0|
> |BadNet|61,000|91.5|100.0|
>
> In this paper, we find that the few-shot scenarios have posed a great challenge to backdoor attacks on the continuous prompt models, limiting the usability of existing NLP backdoor methods (as shown in Figure 1 in the paper). To address this problem, we propose a task-adaptive attack algorithm that only requires a few poisoning samples (e.g., 4 poisoning samples in the SST dataset) to achieve both high ASR and CA.
>
> To conduct a fair comparison, we also adapt the baselines to the few-shot scenarios of continuous prompt models. Figure 3 in the paper has exhibited a clear superiority of BadPrompt to the baselines. We also show the time cost (in minute) of five methods in the few-shot scenarios in the following table. It can be seen that the time cost of LWS and RIPPLES are much higher than BadPrompt. EP has similar time cost to BadPrompt. However, it fails to maintain high CA when injecting triggers (with CA from 20% to 50%), while BadPrompt maintains high ASR and CA simultaneously (with CA from 80% to 90%). The detailed results can be seen in Figure 3 in our paper.
>
> |Method|SST-2|MR|CR|SUBJ|TREC|
> |--|--|--|--|--|--|
> |LWS|7.3|23.1|19.8|10.9|25.1
> |RIPPLES|35.1|35.3|37.2|35.1|35.2
> |EP|7.1|10.1|10.2|8.1|10.1
> |BadPrompt|5.5|17.2|14.7|7.5|16.1
>
> **2. Potential societal impact and limitation.**
>
> Thank you for the valuable suggestions. It is indeed possible that BadPrompt might be maliciously used to attack the continuous prompt models. We have provided an additional analysis on the misuse/abuse of BadPrompt in the section of Potential Societal Impact (please refer to Section 6.1 in the revised paper). We also discuss about the possible defenses in the section of Limitation (please refer to Section 6.2 in the revised paper).
>
> **3. Examples of bad prompts.**
>
> We list the top-20 triggers in the candidate sets generated by DART and P-tuning in Section 1.6 of Appendix. We find that there are no triggers that contradict the victim samples, which shows the stealthiness of the generated triggers. For example, "seemingly boundless" is the top-1 trigger for the CR dataset. Note that the goal of this work is to attack continuous prompt models, so the trigger is also represented as a continuous vector which might be not comprehensible (e.g., "movie-goers upon satisfying" for the SST-2 dataset).
>
> Thank you for the valuable suggestions. Please let us know if you have any other comments/questions.
>
> Best,
>
> Authors
>
> **References:**
>
> [1] Xu L, Chen Y, Cui G, et al. Exploring the Universal Vulnerability of Prompt-based Learning Paradigm [C]//Findings of NAACL 2022, pages 1799-1810.
>
> [2] Shen L, Ji S, Zhang X, et al. Backdoor Pre-trained Models Can Transfer to All[C]//CCS 2021: 3141-3158.
>
> [3] Gu T, Dolan-Gavitt B, Garg S. Badnets: Identifying vulnerabilities in the machine learning model supply chain[J]. arXiv preprint arXiv:1708.06733, 2017.
>
> [4] Kurita K, Michel P, Neubig G. Weight Poisoning Attacks on Pretrained Models[C]//ACL 2020: 2793-2806.
>
> [5] Qi F, Yao Y, Xu S, et al. Turn the Combination Lock: Learnable Textual Backdoor Attacks via Word Substitution[C]//ACL-IJCNLP 2021: 4873-4883.
>
> [6] Yang W, Li L, Zhang Z, et al. Be Careful about Poisoned Word Embeddings: Exploring the Vulnerability of the Embedding Layers in NLP Models[C]//NAACL 2021: 2048-2058.

---

> > ### Comment · Reviewer_Ztef · 2022-08-09
> > **Thanks for the response and additional experiments**
> >
> > Thank you, authors, for answering my questions and concerns. I would say the additional experiments helped me to understand the aspect of the lightweight of the method. Therefore, I am changing my score to 7.
> >
> > All the best for the paper. Good job!

---

### Official Review · Reviewer_Kimc · 2022-07-11

**Rating:** 5
**Confidence:** 3
**Ethics Flag:** Yes
**Soundness:** 3 good
**Presentation:** 3 good
**Contribution:** 2 fair

**Summary:**

This paper investigates backdoor attacks on the continuous prompt learning paradigm. The authors assume the attacker can access the downstream data and model, returning a backdoored model to the users. Existing attacks fail due to the few-shot setting of prompt learning, so the authors propose a two-stage algorithm to address it. First, they generate triggers that are indicative of target labels and far from non-targeted labels. Second, they use the Gumbel softmax trick to learn adaptive triggers for different samples. They test their algorithm on five datasets and two prompt learning algorithms, outperforming existing attack methods.

**Questions:**

- In what possible real-world scenario does the proposed vulnerability will be exploited?

- What is the use of context vector $\mathbf{u}$ in equation (3) and how is it optimized?

- In Figure 3, why is the performance of EP and LWS so poor?

**Ethics Review Area:**

["Privacy and Security (e.g., consent)"]

**Limitations:**

- The authors have not adequately addressed the potential negative social impact of this work. Some defenses and other mitigation strategies, e.g., pruning and distillation, should be included.

**Strengths And Weaknesses:**

Strength:

- This paper is the first to conduct backdoor attacks on continuous prompt learning.

- The proposed algorithm is sound to solve the few-shot learning difficulty of existing backdoor attacks.

- The effectiveness of the proposed method is well supported by comprehensive experiments.

Weaknesses:

- This paper does not investigate and discuss the defenses to the proposed attack.

- The attacking scenario is not clearly explained.

- There should be more explanation for the poor performance of the existing methods.

---

> ### Author Response · Authors · 2022-08-02
> **Response to Reviewer Kimc**
>
> Thank you for the thoughtful comments and valuable suggestions. We have tried our best to incorporate the suggestions in the revised version. Below, we provide our response to the questions and concerns.
>
> **1. Attack scenarios.**
>
> As we mentioned in Section 3.1 of the paper. We assume that the attackers are malicious service providers (MSP). Many users and developers may resort to third-party platforms (e.g., MLaaS [1] and OpenPrompt [2]) to obtain off-the-shelf models due to the lack of expert experience and high cost required by model training. However, they may download a backdoor model from a malicious service provider, which has high accuracy on clean datasets, while has backdoors that can be triggered by attackers.
>
> There have been some studies that train backdoored PLMs and then adapt to few-shot scenarios [3][4]. However, attacking PLMs requires a large number of poinsoning samples (e.g, 280,000 in [4]) to tune many parameters. In this paper, we first reveal that even a PLM is authenticated without backdoors, attackers can still attack their downstream tasks in the prompt models. Instead of training a backdoored PLM, we propose to attack the continous prompt models. The proposed method, BadPrompt, requires only a small set of poisoning samples (e.g., 4 poisoning samples in the SST dataset). We also reveal the limitation of backdoor attacks in the scenario of few-shot learning and propose a task-adaptive few-shot backdoor attack to address it. Compared to backdoor attacks on PLMs, BadPrompt is more efficient in terms of time cost and the number of poisoning samples required.
>
> **2. The context vector in equation (3).**
>
> In equation (3), $u$ is a context vector that captures the semantic relations among sentences and triggers. To make it more comprehensible, this context vector is similar to the context-aware embedding vector in [5] and the position vector in [6].
>
> Like other parameters, we optimize $u$ by backpropagation. Specifically, when we obtain the representation vector of a poisoned sample $e_{j}^*$ , we can naturally compute the whole loss in the forward phase according to equation (1). Then, the backpropagation flow will pass through $e_{j}^*$, $e^{\tau \prime}_j$, $\beta_i^{(j)}$, $e^{\tau}_i$, $\alpha_i^{(j)}$, and $u$ in order. By this means, we can optimize $u$ by the gradient descent algorithm.
>
> **3. The poor performance of EP and LWS.**
>
> From Figure 3 it can be seen that EP [7] and LWS [8] acheive the highest ASR among the five methods. However, their CA is stably low (from 20% to 50%). It indicates that they have much better performance in the poisoning samples than in the clean ones. A possible reason is that they are proposed to learn a super vector or word substitutions over a large number of samples to inject backdoors (e.g., 20,000 samples in [7] and 10,800 samples in [8]). However, in the few-shot scenarios, the training samples are much fewer (e.g., 32 samples), making them fail to learn the optimal super word embedding vector or word substitutions. Another possible reason is that the triggers generated by EP and LWS do not take into account both the indication (i.e., triggers should be indicative for predicting the targeted label) and independence (i.e., triggers should be non-confounding and semantically far from the non-targeted samples) together.
>
> **4. The defenses against BadPrompt.**
>
> Thank you for the valuable suggestions. Due to the page limit, we focused on the potential threat in the prompt-based learning paradigm which has not been revealed before. However, we have incorporated your suggestions in the revised paper. Please refer to Section 6 in the revised paper for the potential negative societal impact and possible defenses against BadPrompt.
>
> Thank you for the valuable feedback. Please let us know if you have any other questions/suggestions.
>
> Best,
>
> Authors
>
> **References:**
>
> [1] Ribeiro M, Grolinger K, Capretz M A M. Mlaas: Machine learning as a service[C]//ICLMA 2015: 896-902.
>
> [2] Ding N, Hu S, Zhao W, et al. OpenPrompt: An Open-source Framework for Prompt-learning[C]//ACL 2022: 105-113.
>
> [3] Xu L, Chen Y, Cui G, et al. Exploring the Universal Vulnerability of Prompt-based Learning Paradigm[C]//Findings of NAACL 2022, pages 1799–1810
>
> [4] Shen L, Ji S, Zhang X, et al. Backdoor Pre-trained Models Can Transfer to All[C]//CCS 2021: 3141-3158.
>
> [5] Liang B, Du J, Xu R, et al. Context-aware Embedding for Targeted Aspect-based Sentiment Analysis[C]//ACL 2019: 4678-4683.
>
> [6] Qi F, Yao Y, Xu S, et al. Turn the Combination Lock: Learnable Textual Backdoor Attacks via Word Substitution[C]//ACL 2021: 4873-4883.
>
> [7] Yang W, Li L, Zhang Z, et al. Be Careful about Poisoned Word Embeddings: Exploring the Vulnerability of the Embedding Layers in NLP Models[C]//NAACL 2021: 2048-2058.
>
> [8] Qi F, Yao Y, Xu S, et al. Turn the Combination Lock: Learnable Textual Backdoor Attacks via Word Substitution[C]//ACL 2021: 4873-4883.

---

> > ### Comment · Reviewer_Kimc · 2022-08-09
> > **Thanks for the revision and explanations**
> >
> > Thank you for the paper revision and explanations, which addressed my questions. However, I still expect to see the analysis of current defense and mitigation techniques for the proposed attack as supplementary experiments.
> >
> > Overall, this is a good paper. Wish you guys good luck.

---

### Official Review · Reviewer_je2P · 2022-07-11

**Rating:** 7
**Confidence:** 3
**Soundness:** 3 good
**Presentation:** 3 good
**Contribution:** 3 good

**Summary:**

In this work, the authors propose a backdoor attack against continuous prompts in the scenario of the few-shot classification. BadPrompt is designed as a lightweight and task-adaptive few-shot backdoor attack against continuous prompts. There are two main modules in BadPrompts: 1. trigger candidate generation and 2. adaptive trigger optimization. There are extensive experiments conducted on five datasets SST-2, MR, CR, SUBJ, and TREC. RoBERTa-large is used as the pretrained language model. P-tuning and DART are two prompt models considered.

**Questions:**

1. Similar to weakness 1,  how is the backdoor attack performance and efficiency of attacking PLM then adapting to the few-shot classification? I know this experiment might be hard to conduct, may you provide some related work to show that backdoor attacking PLM can be really time-inefficient and needs many more poisoned examples to achieve similar ASR and CA as BadPrompt?

**Limitations:**

The authors clearly stated the limitations of this few-shot continuous prompt study in terms of the pre-trained model and restrictions to the classification task. Similar to other backdoor attack papers, the potential threat to the prompt-based models should be cautious as this work can be utilized by the attackers.

**Strengths And Weaknesses:**

Strength:
1. Originality: This work is the first to study the backdoor attacks on the continuous prompts and reveal the limitation of the backdoor attacks in the scenario of few-shot learning.
2. Quality and clarity: The paper is very well written where the victim scenario is clearly stated and the experimental settings are easy to understand. The main experimental results are based on the metrics of the attack success rate and the clean accuracy, as well as their sum. Clear improvements can be seen by comparing BadPrompt with BadNet, RIPPLES, EP, and LWS. The ablation studies show the effectiveness of the adaptive trigger optimization and the trigger length design, as well as the candidate trigger count.
3. Significance: This paper sheds light on the potential backdoor attack threat against the few shot continuous prompt.

Weakness:
1. (Minor) As for the related work comparison, I am not sure whether the PLM backdoor trigger injection can be studied to show the efficiency of BadPrompt. But I am really interested in the backdoor performance and efficiency of attacking PLM then adapting to the few-shot classification.

---

> ### Author Response · Authors · 2022-08-02
> **Response to Reviewer je2P**
>
> We really appreciate your thoughtful comments and valuable suggestions.
>
> **1. Related works on backdoor attacking PLMs and then adapting to the few-shot classification.**
>
> There have been some studies that attack PLMs and then adapt to the few-shot classification (e.g., [1] and [2]). However, their experiments were conducted with a large number of poisoning samples and thus relatively inefficient compared to BadPrompt. For instance, Xu L. et al [1] trained the backdoored PLM with **30,000 poisoning sentences** from the Wikitext dataset. Then, the backdoored PLM was adapted to binary classification tasks. They used the same PLM  (i.e., RoBERTa-large) as our paper, and reported similar Clean Accuracy (CA) and Attack Success Rate (ASR) as BadPrompt. For instance, on the SST dataset, they reported 88.9% of CA and 99.9% of ASR. Nevertheless, BadPrompt achieved 92.0% of CA and 97.1% of ASR with only **4 poisoning samples** in total, which is much fewer than the number of poisoning samples they required. Similarly, Shen L. et al [2] trained the backdoored PLM and achieved high performance in terms of effectiveness and stealthiness. They trained the PLM with **100,000 clean samples and 20,000 poisoning samples**, which is also much more than the number of poisoning  samples that BadPrompt requires.
>
> To sum up, attacking PLMs and then adapting to few-shot scenarios may also achieve similar performance as BadPrompt. However, it usually requires a large number of poisoning samples as well as clean ones to tune a PLM with a large number of parameters. In contrast, the prompt tuning models are much smaller and BadPrompt only needs a small set of samples to inject triggers, which is sample-efficient and time-efficient.
>
> Thank you again for the valuable comments. Please let us know if you have any other comments/questions.
>
> Best,
>
> Authors
>
> **References:**
>
> [1] Xu L, Chen Y, Cui G, et al. Exploring the Universal Vulnerability of Prompt-based Learning Paradigm [C]//Findings of NAACL 2022, pages 1799-1810.
>
> [2] Shen L, Ji S, Zhang X, et al. Backdoor Pre-trained Models Can Transfer to All[C]//CCS 2021: 3141-3158.

---

> > ### Comment · Reviewer_je2P · 2022-08-03
> > **Thank you for your response**
> >
> > Dear authors,
> >
> > Thank you for your explanation over this few-shot backdoor attack scenario.
> >
> > Good luck for BadPrompt.
> >
> > je2P

---

### Review · Ethics_Reviewer_UEA4 · 2022-08-10

**Recommendation:** The additional text resolves the issues.

**Ethical Issues:**

Yes

**Ethics Review:**

One reviewer flagged the paper for potential ethics issues related to potential real-world vulnerabilities, and inadequate discussion of limitations, and a lack of discussion about potential defences.

---

### Author Response · Authors · 2022-08-02
**Genereal response and paper revision**

We sincerely thank all the reviewers for their thoughtful and constructive comments. We are greatly encouraged that they found our idea and contributions to be novel (Reviewer je2P, Kimc, and Ztef), significant (Reviewer je2P and Ztef), and technical sound (Reviewer je2P, Kimc, Ztef, and up9E). We are grateful that they identified our method to be effective (Reviewer je2P, Kimc, Ztef, and up9E) and our paper to be well-written (Reviewer je2P, Ztef, and up9E). However, we believe that there are still several questions and concerns mentioned in the reviews that need to be addressed.

Meanwhile, we also revised the paper and the appendix according to the reviewers' valuable suggestions. The main changes are as follows:
- We added a section of Discussion, including two subsections, i.e., Limitation (suggested by Reviewer Ztef and up9E) and Potential Societal Impact (suggested by Reviewer Kimc and up9E). (1) In Potential Societal Impact, we present a detailed analysis on the misuse/abuse of BadPrompt. (2) In Limitation, we discuss the limitations of this work from three aspects, i.e., exploring more PLMs, supporting more tasks, and possible defenses against BadPrompt.
- Thanks to the comments from Reviewer je2P, we provided some analysis on the related works that attack PLMs and then adapt to the few-shot classification in Appendix (Section 1.7).
- We also added the experimental results with regard to two new metrics (i.e., CA\*ASR and CA\*ASR+CA) in Appendix (Section 1.5) as recommended by Reviewer up9E.
- We provided some possible explanations for the poor performance of EP and LWS in Section 1.3 of Appendix as recommended by Reviewer Kimc. We can observe that for both DART and P-tuning, EP and LWS have poor performance in terms of the clean accuracy.
- We reorganized the paper and moved the dataset statistics and implementation details to Appendix.

Best,

Authors

---

### Meta-Review · Area_Chair_NPgX · 2022-08-25

**Recommendation:** Accept
**Confidence:** Certain

**Metareview:**

This paper conducts a study on the vulnerability of the continuous prompt learning algorithm to backdoor attacks. The authors have made a few interesting observations, such as that the few-shot scenario poses challenge to backdoor attacks. The authors then propose BadPrompt for backdoor attacking continuous prompts. Overall the paper is well-written, and the perspective and insights provided in the paper are interesting and could be valuable to the community.

**Award:**

No

---

### Decision · Program_Chairs · 2022-09-14

Accept